



**Differentiation of cognate bacterial communities in thermokarst**
**landscapes: implications for ecological consequences of permafrost**
**degradation**
Running title: Bacterial communities in thermokarst landscape
**Ze Ren[1,2]\*, Shudan Ye[3], Hongxuan Li[3], Xilei Huang[3], Luyao Chen[3]**
1    State Key Laboratory of Lake Science and Environment, Nanjing Institute of Geography and Limnology,
Chinese Academy of Sciences, Nanjing 210008, China
2    Advanced Institute of Natural Sciences, Beijing Normal University, Zhuhai, 519087, China
3    Faculty of Arts and Sciences, Beijing Normal University, Zhuhai 519087, China
**\*Corresponding Authors**:
Ze Ren: renzedyk@gmail.com
**Emails:**
Shudan Ye: yeshudan@mail.bnu.edu.cn; Hongxuan Li: lihongxuan@mail.bnu.edu.cn; Xilei Huang:
202111079031@mail.bnu.edu.cn; Luyao Chen: 202011059371@mail.bnu.edu.cn



**Abstract**
Thermokarst processes likely result in new habitats harboring novel bacterial communities
in degraded permafrost soil (PBCs), thermokarst lake sediments (SBCs), and lake water
(WBCs). Our study aimed to investigate the paired PBCs, SBCs, and WBCs across the
Qinghai-Tibet Plateau (QTP) by assessing the spatial pattern of diversity as well as
assembly mechanisms of these bacterial communities. Each habitat had distinct bacterial
assemblages, with lower alpha diversity and higher beta diversity in WBCs than in SBCs
and PBCs. However, up to 41% of the OTUs were shared by PBCs, SBCs, and WBCs,
suggesting that many taxa originate from the same sources via dispersal. SBCs and WBCs
had reciprocal dispersal effects and both were correlated with PBCs. Dispersal limitation
was the most dominant assembly process shaping PBCs and SBCs while homogeneous
selection was the most dominant for WBCs. Bacterial communities of the three habitats
correlated differently with environmental variables, but latitude, mean annual precipitation,
and pH were the common factors associated with their beta diversity, while total
phosphorus was the common factor associated with their assembly processes. Our results
imply that thermokarst processes result in diverse habitats that have distinct bacterial
communities that differ in diversity, assembly mechanisms, and environmental drivers.
**Keywords:** thermokarst; permafrost; bacteria, community assembly, Qinghai-Tibet
Plateau
**1 Introduction**
Permafrost is an important landscape in high latitude and altitude regions, covering 15%
of the land area of the Northern Hemisphere (Obu, 2021) and 40% of the Qinghai-Tibet
Plateau (QTP) (Zou et al., 2017; Gao et al., 2021), and containing twice as much carbon as



is currently present in the atmosphere (Schuur et al., 2009; Hugelius et al., 2014; Mishra et
al., 2021). Permafrost is highly sensitive to climate warming (Wu et al., 2007; Jorgenson
et al., 2010; Biskaborn et al., 2019), which is expected to reduce 50-90% of permafrost
cover by 2100 (Lawrence et al., 2012; Chadburn et al., 2017). As a result of ice-rich
permafrost thaw, thermokarst lakes and ponds are formed (Kokelj and Jorgenson, 2013;
Farquharson et al., 2016) and extensively distributed across the Arctic and sub-Arctic
regions (de Jong et al., 2018) as well as the QTP (Niu et al., 2011; Luo et al., 2020). The
initial sediment and water in thermokarst lakes originate from the melting of permafrost,
and they are continuously replenished through the collapse of permafrost and precipitation
(West and Plug, 2008; de Jong et al., 2018). Thus, thermokarst lake sediments and water,
as well as the surrounding degraded permafrost soil, represent three distinct habitats
derived from the original permafrost during the process of thermokarst formation (Figure
1). It is well known that thermokarst processes substantially influence regional
hydrological, ecological, and biogeochemical processes (Chin et al., 2016; In'T Zandt et
al., 2020; Manasypov et al., 2021) and initiate a strong positive climate feedback to global
warming (Walter et al., 2006; Schuur et al., 2008; Schaefer et al., 2011; Anthony et al.,
2018). However, the microbial differences and relationships among these distinct habitats
in thermokarst landscapes are largely unknown.
Understanding microbes in thermokarst landscapes, and elsewhere, is important because
microbial communities play pivotal roles in driving biogeochemical and ecological
processes. To understand thermokarst microbial communities, we need to understand the
assembly mechanisms structuring them, a central research topic in microbial ecology
(Stegen et al., 2012; Nemergut et al., 2013; Zhou et al., 2014; Zhou and Ning, 2017). In



the assembly of microbial communities, both deterministic and stochastic processes occur
simultaneously but with contributions that can vary (Chase, 2010; Zhou et al., 2013;
Vellend et al., 2014; Makoto et al., 2019). Typically, deterministic processes place a strong
emphasis on niche-based mechanisms, including ecological selection, environmental
filtering, and biotic interactions (Zhou and Ning, 2017). Conversely, stochastic processes
involve neutral mechanisms like random birth and death, unforeseen disturbance,
probability-based dispersal, and ecological drift (Chave, 2004; Chase, 2010; Zhou et al.,
2014). In various ecosystems or habitats, the significance of deterministic and stochastic
processes can differ greatly and be shaped by a multitude of environmental factors (Tripathi
et al., 2018; Aguilar and Sommaruga, 2020; Jiao and Lu, 2020; She et al., 2021). During
thermokarst formation, vast areas of permafrost have been transformed to thermokarst
lakes, leading to major changes in physicochemical environments as well as in biological
communities of these regions. Thus, it is also expected that the microbial communities
experience major changes in occupying degraded permafrost soil, thermokarst lake
sediments, and lake water, and in doing so, display different assembly mechanisms (Figure

77   1).

Better understanding community assembly in these systems is important because thawing
permafrost and thermokarst lakes are greenhouse gas emission hotspots (In'T Zandt et al.,
2020; Mu et al., 2020; Elder et al., 2021). Close relationships between biogeochemical
processes and microbial community assembly have been generally demonstrated (Bier et
al., 2015; Graham et al., 2016; Le Moigne et al., 2020; Ren et al., 2022a). Assembly
processes inevitably influence biogeochemical functions by shaping community diversity
and composition (Graham et al., 2016; Leibold et al., 2017; Mori et al., 2018). For example,



dispersal (a stochastic process) can suppress biogeochemical functioning by increasing the
proportion of maladapted taxa (Strickland et al., 2009; Nemergut et al., 2013; Graham and
Stegen, 2017), while selection (a deterministic process) may have positive influence on
biogeochemical function by facilitating locally adapted taxa (Graham et al., 2016). In
particular, stochastic dispersal has been suggested to suppress the mineralization of organic
carbon in soil and water (Le Moigne et al., 2020; Luan et al., 2020). Therefore, it is
hypothesized that the relative influence of deterministic and stochastic processes on
community assembly could impact the biogeochemical functions of microbial
communities (Strickland et al., 2009; Nemergut et al., 2013; Pholchan et al., 2013; Graham
and Stegen, 2017). Given the importance to understand how microbial community
variations affect the biogeochemical cycles in permafrost and thermokarst landscapes, it is
necessary to have a deeper understanding of the assembly mechanisms in shaping
microbial communities that form following permafrost degradation.
In this paper we evaluated these ideas on the Qinghai-Tibet Plateau (QTP), which is known
as the "Third Pole" of the Earth and is therefore uniquely positioned as an indicator of
global change (Yao et al., 2012). Pronounced environmental changes in response to climate
warming on the QTP have been observed and documented, especially in the past half
century (Piao et al., 2012; Zhang et al., 2018; Ren et al., 2019a). Major changes are
predicted to continue on the QTP and permafrost thawing is among the most prominent but
little is known about the microbial communities in these rapidly emerging ecosystems. To
fill this gap, we investigated water and sediment in thermokarst lakes across the QTP as
well as permafrost soil around the lakes (Figure 1). Our aims were to (1) assess the spatial
pattern of alpha and beta diversity of bacterial communities, and (2) evaluate the



community assembly mechanisms and environmental responses of the bacterial
communities in degraded permafrost soil, as well as in the sediment and water of
thermokarst lakes.
**2 Methods**
***2.1 Study area, field sampling, and chemical analysis***
This work was conducted across the QTP in July 2021 (Figure S1). In total, 44 sites were
investigated by collecting paired samples of lake water, lake sediment, and surrounding
permafrost soil (Figure 1a) (Ren et al., 2022a). The sampling strategy and chemical
analysis methods were described in detail in our previous publications (Ren et al., 2022a,
b). For water sampling of each lake, surface water samples were collected at a depth of 0.3
to 0.5 m with three replicates. For microbial analysis, 200 mL of water was filtered using
a 0.2-μm polycarbonate membrane filter (Whatman, UK) for DNA extraction. The
remaining water samples were transported to the lab for other physicochemical
measurements. For sediment sampling, the top 15 cm of sediment was collected from 3
points. Sediment samples for microbial analysis were collected in a 45-mL sterile
centrifuge tube, and the remaining samples were air-dried for analyzing physicochemical
properties. For permafrost sampling, five topsoil cores were collected along three 25-m
transects with increasing distances to the lake shore, respectively. The soils from one
transects were homogenized. Soil samples for microbial analysis were stored in 45-mL
sterile centrifuge tubes and the remaining soils were used for analyzing physicochemical
properties. For each sampling site, pH, conductivity (Cond), organic carbon (DOC in water
and SOC in sediment and soil), total nitrogen (TN), and total phosphorus (TP) were
measured according to our previous publications (Ren et al., 2022a, b). Moreover, the QTP





climate dataset (Zhou, 2018) was obtained from the National Tibetan Plateau Data Center
(https://data.tpdc.ac.cn/en/), and was utilized to extract the mean annual temperature (MAT)
and mean annual precipitation (MAP) for each of the study sites.

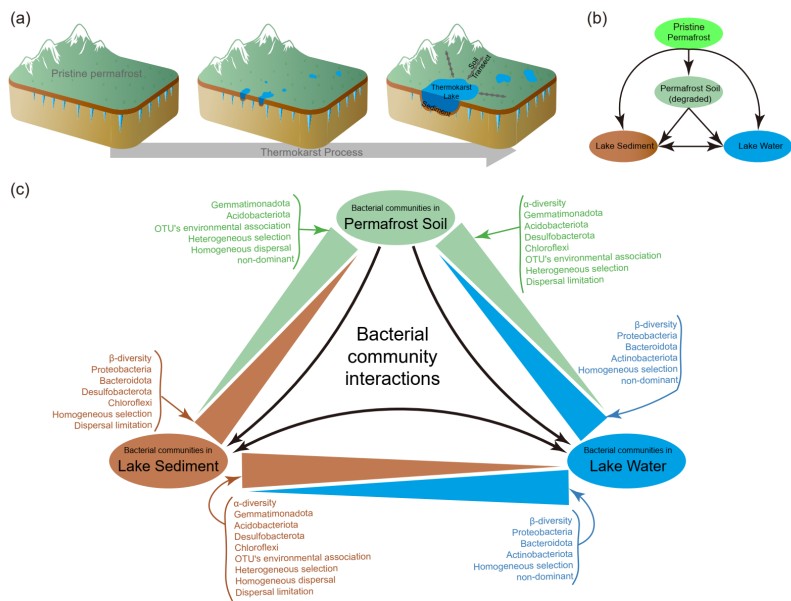


Figure 1 (a) The process of thermokarst lake formation in ice-rich permafrost (modified
from Ren et al, 2022a). (b) A schematic view of the relationships between permafrost soil,
lake sediment, and lake water. (c) Summary of the differences between distinct habitats of
the bacterial communities in permafrost soil, lake sediment, and lake water.
***2.2 DNA extraction, PCR, and sequencing***
The methods of DNA extraction, PCR, and sequencing were described in detail in our
previous publication (Ren et al., 2022a). In brief, the Magen Hipure Soil DNA Kit (Magen,
China) was used to extract DNA from soil, sediment, and water samples according to the
manufacturer's protocols. The prokaryotic 16S rRNA gene's V3-V4 hypervariable regions
were amplified using universal primers 343F-TACGGRAGGCAGCAG and 798R-



AGGGTATCTAATCCT (Nossa et al., 2010). To reduce amplification bias, three
individual PCR amplifications were performed for each sample and the triplicate PCR
products were combined, purified, and quantified. Sequencing of the amplicon products
was done on the Illumina MiSeq platform (Illumina, San Diego, CA, USA) following the
manufacturer's instructions. Raw sequences were trimmed of ambiguous bases and low-
quality sequences, and paired-end reads were joined and de-noised using QIIME1.9.1
(Caporaso et al., 2010). The effective sequences were grouped into Operational Taxonomic
Units (OTUs) using a 97% sequence similarity threshold against the SILVA 138 database
(Quast et al., 2013). The singletons were removed, and the sequences were normalized to
24,251 sequences per sample to eliminate the bias from the sampling effort.
*2.3 Analyses*
Three α-diversity indices, including observed number of OTUs (OTU richness), Shannon
diversity, and phylogenetic diversity (PD), were calculated using QIIME 1.9.1 (Caporaso
et al., 2010). The "ses.mntd" function in the picante 1.8.2 package was used to calculate
the standardized effect size measure of the mean nearest taxon distance (SES.MNTD) for
assessing the phylogenetic clustering of bacterial communities (Kembel et al., 2010). The
β-diversity was calculated as the Bray-Curtis distance based on the relative abundance of
OTUs. Differences in α-diversity and β-diversity among bacterial communities in different
habitats, including permafrost soil bacterial communities (PBCs), lake sediment bacterial
communities (SBCs), and lake water bacterial communities (WBCs), were assessed using
Wilcoxon rank-sum test. The relationships between taxonomic and environmental
variables were assessed using Spearman correlation. Mantel tests were performed to
examine the correlation between environmental variables and β-diversity. A Non-metric





Multidimensional Scaling (NMDS) analysis was conducted to examine the distribution of
PBCs, SBCs, and WBCs using the "metaMDS" function in the vegan 2.5-7 package
(Oksanen et al., 2020). The distinctiveness of these communities was confirmed through a
non-parametric statistical test (ANOSIM) using the "anosim" function in the vegan
package. The habitat niche occupied by each species was estimated by calculating Levin's
niche breadth (Levins, 1968) with the use of the spa 0.2.2 package (Zhang, 2016). Species
with a broader niche breadth were distributed more evenly across a wider range of habitats,
compared to those with a narrower niche breadth.
Structural equation modeling (SEM) was conducted to assess the relationships among
location (including latitude, longitude, and elevation), climate (including mean annual
temperature and mean annual precipitation), and physicochemical parameters (including
pH, conductivity, nutrients concentrations and stoichiometric ratios) of each habitat. In the
SEM, location, climate, and physicochemical environments were reduced in dimensions
by principal component analysis (PCA), respectively, using the "prcomp" function of the
vegan package, and the first axis (PCA1) was used. For community structure, the first axis
of NMDS was used.
Phylogenetic trees of bacteria were constructed in the R package ggtree 3.2.1 (Yu et al.,
2017) using the top 1000 abundant OTUs in PBCs, SBCs, and WBCs, respectively. For
each phylogenetic tree, a heatmap was built in the inner ring represents Spearman's
correlation between OTUs and environmental variables. The middle ring was built to
represent the frequency of the OTUs in our studied sites. The outer ring was built to
represent the relative abundance of the OTUs.




A null model analysis was performed to investigate the processes shaping the assembly of
bacterial communities in permafrost soil, lake sediment, and lake water using the R
package picante 1.8.2 (Kembel et al., 2010). This analysis based on the calculation of the
beta nearest taxon index (βNTI) to measure the extent of deterministic processes in shaping
the phylogenetic composition of the communities, as well as a Bray–Curtis-based Raup-
Crick matrix (RC$_{Bray}$) to assess the relative influences of stochastic processes (Stegen et al.,
2013; Zhou and Ning, 2017). Mantel tests were conducted to test the relationships between
environmental variables and βNTI.
All the statistical analyses were carried out in R 4.1.2 (R Core Team, 2020).
**3 Results**
*3.1 General distribution patterns of α-diversity*
After quality filtering, 3,201,132 high quality sequences were obtained and clustered into
9,361 OTUs, of which, 3870 OTUs were core OTUs shared by bacterial communities in
permafrost soil, lake sediment, and lake water (Figure S2). Moreover, a large number of
OTUs were shared by PBCs and SBCs (n=7053), of which, 16.4% were enriched in lake
sediment and 19.3% were enriched in permafrost soil (Figure S2). However, a relatively
small number of OTUs were shared by PBCs and WBCs (n=4007) and by SBCs and WBCs
(n=4431), and only a very small proportion of OTUs were enriched in lake water (Figure
S2). Bacterial communities had a significantly lower α-diversity in lake water than in lake
sediment and permafrost soil (Figure 2a). α-diversity was not significantly different
between PBCs and SBCs (Figure 2a). Correlation analyses showed that phylogenetic
diversity of PBCs was positively correlated with SOC, TN, and C:N:P ratios (Figure 2b).
For SBCs, α-diversity indices were positively correlated with MAT, MAP, SOC, TN, C:P



and N:P, while negatively correlated with latitude and pH (Figure 2b). For WBCs, α-
diversity indices were negatively correlated with pH, and Shannon diversity was negatively
correlated with DOC, TN, and C:P (Figure 2b).
PBCs and SBCs had a significantly greater phylogenetic diversity than WBCs (Figure 2a
and Figure 3). The OTUs in PBCs had significantly higher frequency than that of SBCs
and WBCs (Figure 3). The top 1000 abundant OTUs in PBCs were highly correlated with
environmental variables, particularly with latitude, MAP, SOC, TN, TP, and C:N:P ratios
(Figure 3a). The top 1000 abundant OTUs in SBCs were more commonly positively
correlated with MAP, SOC, TN, and C:N:P ratios, but more commonly negatively
correlated with latitude and pH (Figure 3b). The top 1000 abundant OTUs in WBCs had
relatively fewer significant relationships with environmental variables in general, but were
negatively correlated with latitude, conductivity, pH, DOC, TN, and C:N:P ratios, while
more positively correlated with MAP (Figure 3c). In addition, WBCs had significantly
higher SES.MNTD than PBCs and SBCs (Figure 3d), suggesting higher phylogenetic
clustering of bacterial taxa in WBCs.

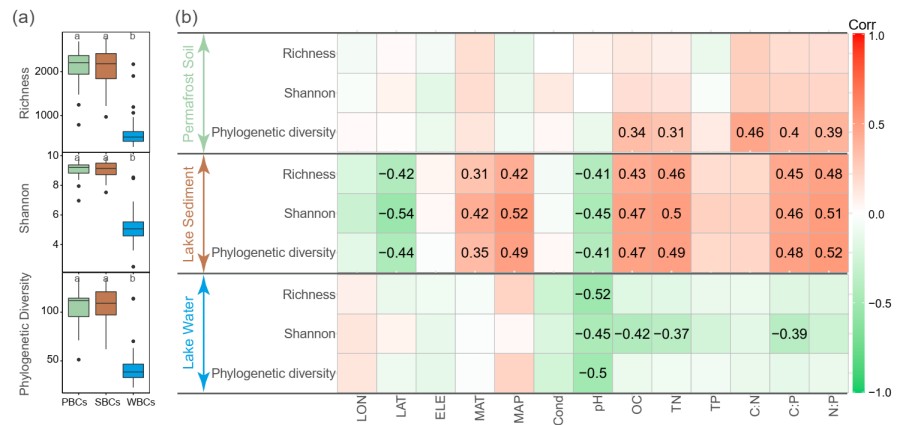




Figure 2 (a) Alpha diversity of bacterial communities in permafrost soil (PBC), lake
sediment (SBC), and lake water (WBC). The different low-case letters represent significant
differences assessed using Wilcoxon rank-sum test. (b) Spearman correlations show the
relationships between alpha diversity and environmental factors. The color represents the
correlation coefficient, which is shown in number when the result is statistically significant
($p<0.05$).

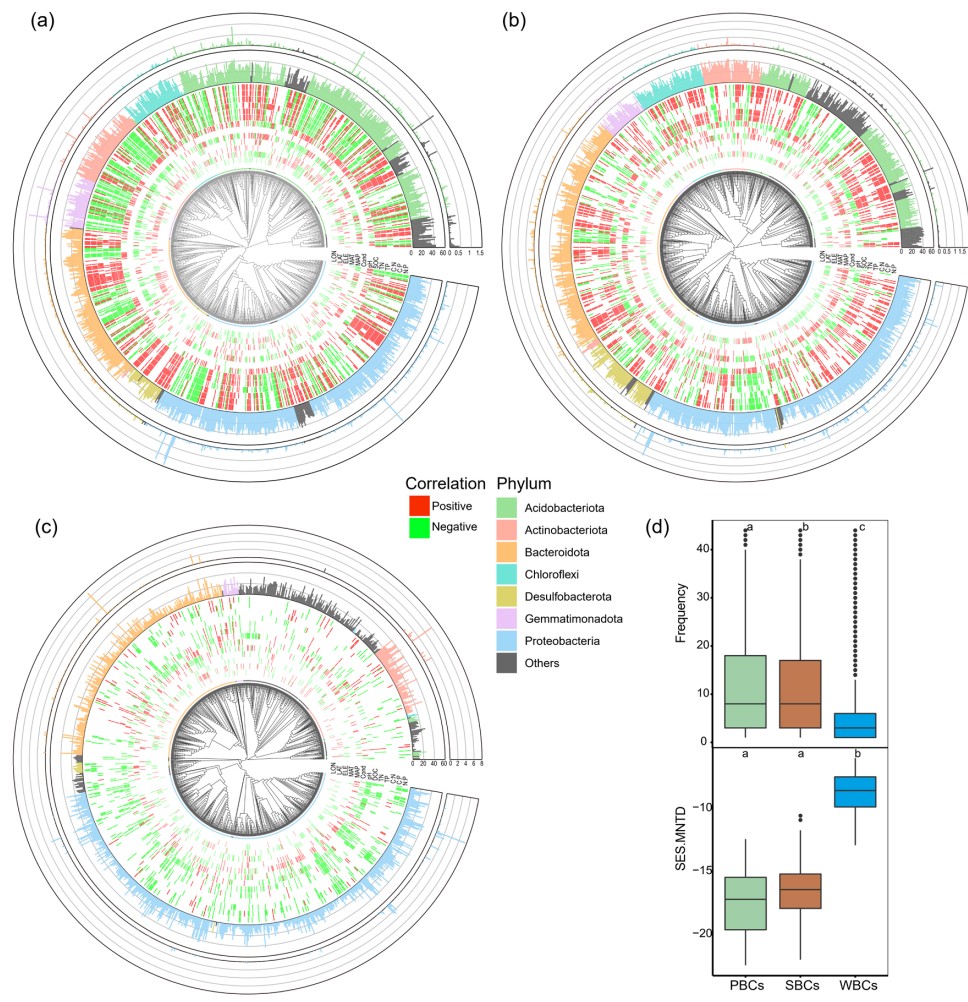




Figure 3 Phylogenetic tree of the top 1000 OTUs in (a) permafrost soil (PBC), (b) lake
sediment (SBC), and (c) lake water (WBC). Tree tips are colored by major phylum. The
inner ring of the heatmap represents spearman's correlation between OTUs and
environmental variables. Only significant (p<0.05) results are shown. The middle ring
represents the frequency of the OTUs in our studied sites. The outer ring represents the
relative abundance of the OTUs. (d) Boxplots showing differences of OTU's frequency and
SES.MNTD values among bacterial communities in permafrost soil (PBCs), lake sediment
(SBCs), and lake water (WBCs). The different lower-case letters represent significant
differences assessed using Wilcoxon rank-sum test.
***3.2 Community composition and β-diversity patterns***
PBCs were dominated by Proteobacteria (30.4%), Acidobacteriota (25.3%), Bacteroidota
(11.4%), Actinobacteriota (6.8%), Chloroflexi (5.2%), and Gemmatimonadota (5.2%)
(Figure 4a). SBCs were dominated by Proteobacteria (35.2%), Bacteroidota (20.0%),
Acidobacteriota (11.3%), Desulfobacterota (6.4%), Chloroflexi (6.3%), and
Actinobacteriota (5.8%) (Figure 4a). WBCs were dominated by Proteobacteria (46.9%),
Bacteroidota (29.2%), and Actinobacteriota (17.4%) (Figure 4a). While Proteobacteria
were predominant in all three habitat types, these dominant phyla had significantly
different relative abundances among these habitats. Proteobacteria and Bacteroidota had a
significantly higher relative abundance in WBCs than in SBCs and PBCs (Figure 4a). The
relative abundance of Actinobacteriota was the highest in WBCs but was not significantly
different between PBCs and SBCs (Figure 4a). Gemmatimonadota and Acidobacteriota
were significantly enriched in PBCs than in SBCs and WBCs. Desulfobacterota and
Chloroflexi were significantly enriched in SBCs than in PBCs and WBCs (Figure 4a).





These phyla responded differently to environmental variables (Figure 4b). For example,
Actinobacteriota and Gemmatimonadota in PBCs and Actinobacteriota and
Desulfobacterota in SBCs were negatively correlated with nutrient concentrations and
ratios, while Desulfobacterota in PBCs and Acidobacteriota in SBCs were positively
correlated with nutrient concentrations and ratios (Figure 4b). pH was a frequent correlate
for taxa in various taxonomic groups across all three habitats (Figure 5b)
Nonmetric multidimensional scaling (NMDS) analysis along with non-parametric
statistical tests showed that bacteria in different habitats formed distinct communities
(Figure 5a). The extent of difference was larger for WBCs vs PBCs ($\beta$=0.98; $R_{ANOSIM}$ =
0.989, P<0.001) than the differences for WBCs vs SBCs ($\beta$=0.96; $R_{ANOSIM}$ = 0.967,
P<0.001). There was the least dissimilarity between PBCs and SBCs ($\beta$=0.81; $R_{ANOSIM}$ =
0.384, P<0.001). The fitted SEM model showed that PBCs had direct effects on SBCs and
WBCs, and the latter two had reciprocal effects on each other (Figure 5b). In addition,
location, climate, and permafrost soil physicochemical environments had direct effects on
PBCs. Climate had direct effects on SBCs while lake water physicochemical environments
had direct effects on WBCs (Figure 5b).
WBCs had a higher $\beta$-diversity than SBCs and PBCs, suggesting that bacterial
communities were more spatially heterogeneous in lake water than in lake sediment and
permafrost soil (Figure 6a). Taxa in PBCs had higher habitat niche breadths than taxa in
SBCs and WBCs (Figure 6b). We estimated the distance decay relationship of bacterial
community similarity. Significant distance-decay relationships were observed for all
communities but the fitness values were relatively low (Figure S3), indicating weak decay
of community similarity with geographic distance in thermokarst landscape. We also




explored the main environmental variables that influence the variations of the bacterial
communities (Figure 7). β-diversities of PBCs, SBCs, and WBCs were all significantly
correlated with latitude, MAP, and pH (Figure 7). In addition, β-diversity of PBCs was also
significantly correlated with all the other environmental variables except MAT and
conductivity. β-diversity of SBCs was also significantly correlated with conductivity and
C:N (Figure 7). β-diversity of WBCs was also significantly correlated with elevation, MAT,
conductivity, DOC, TN, and TP (Figure 7). The results suggested that the compositional
variation among PBCs, SBCs, and WBCs was differentially structured by spatial, climatic,
and physicochemical variables.

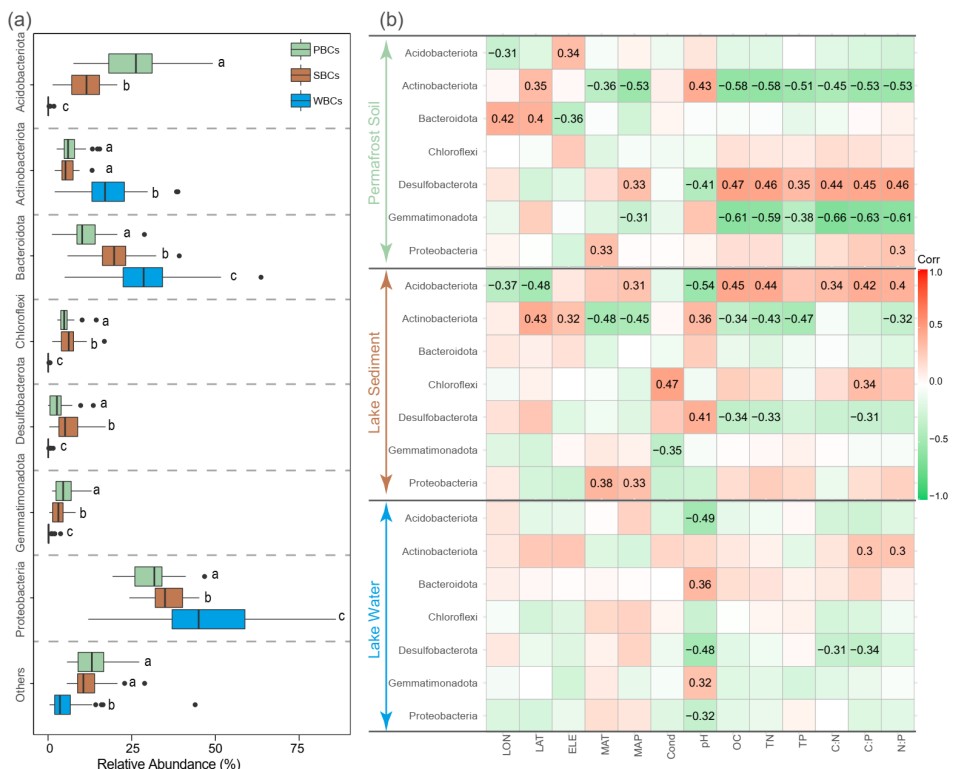






Figure 4 (a) Relative abundances of major phyla in bacterial communities in permafrost
soil (PBCs), lake sediment (SBCs), and lake water (WBCs). The different low-case letters
represent significant differences assessed using Wilcoxon rank-sum test. (b) Spearman
correlations show the relationships between the relative abundance of major phyla and
environmental factors. The color represents the correlation coefficient, which shown in
number when the result is significant (p<0.05).

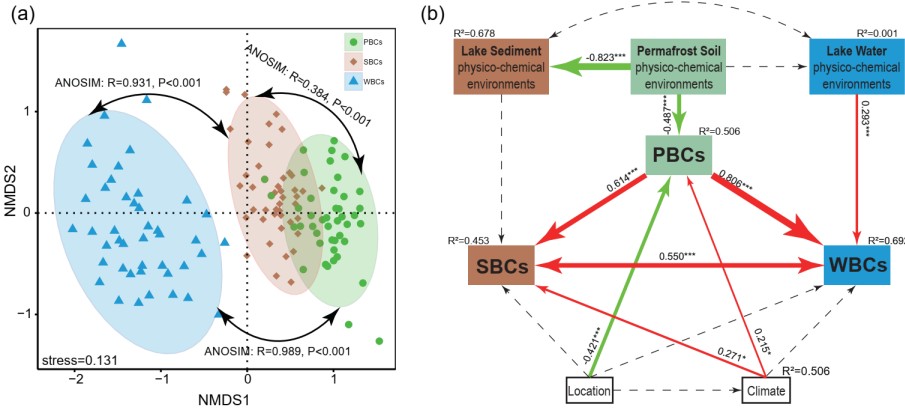


Figure 5 (a) Non-metric multidimensional scaling (NMDS) ordination showing the
distribution of bacterial communities in permafrost soil (PBCs), lake sediment (SBCs), and
lake water (WBCs). The differences between these communities are confirmed by the non-
parametric statistical test (ANOSIM). (b) Structural equation modeling analysis depicting
the relationships between location (including latitude, longitude, and elevation), climate
(including mean annual temperature and mean annual precipitation), physicochemical
environments (pH, conductivity, nutrients concentrations and stoichiometric ratios) of each
habitat. Solid and dashed arrows represent the significant and nonsignificant relationships,
respectively. Red and green arrows represent positive and negative relationships,



respectively. Significant path coefficients are shown adjacent to the path with *, **, and
*** denoting the significant level of $p < 0.05$, $p < 0.01$, and $p < 0.001$, respectively.

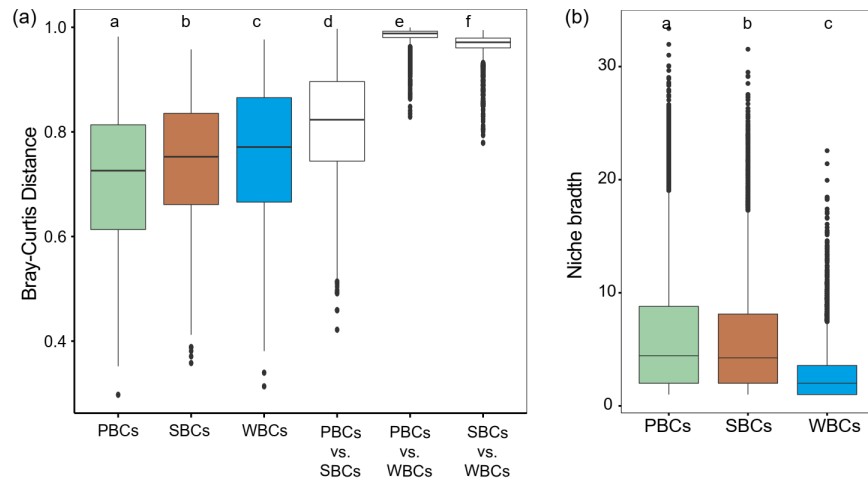


Figure 6 (a) β-diversities within and between PBCs, SBCs and WBCs. (b) Habitat niche
breadth of the bacterial communities.



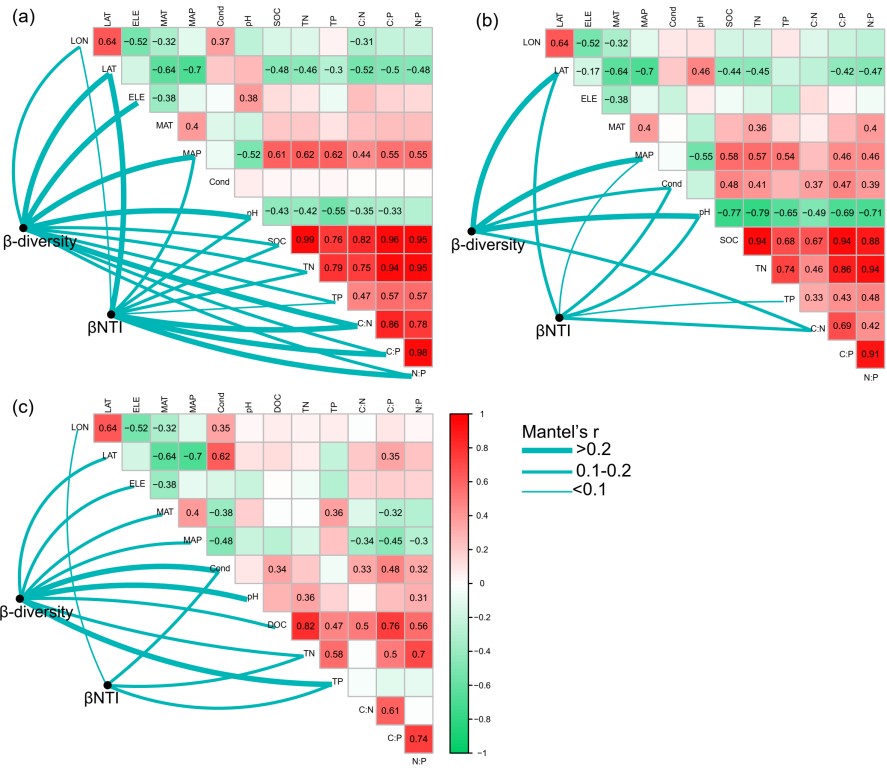

Figure 7 Pairwise correlations between environmental variables as well as the Mantel tests between environmental variables and β-diversity and beta nearest taxon index (βNTI) for (a) bacterial communities in permafrost soil, (b) bacterial communities in lake sediment, and (c) bacterial communities in lake water. β-diversity was calculated as Bray-Curtis distance. The lines denote significant relationships while the line width represents the Mantel's r statistic. Pairwise correlations between environmental variables are shown in color gradient matrix. The color represents Pearson's correlation coefficient, which shown in number when the result is significant (p<0.05). The abbreviations of the environmental variables are explained in the Methods section.



*3.3 Assembly processes*
To explore the mechanisms underlining the observed distribution patterns, a null-model-
based framework was employed to quantify the deviation of phylogenetic turnover. PBCs
had significantly higher βNTI than SBCs and WBCs (Figure 8a). Deterministic processes
contributed 51.3%, 41.2%, and 44.9% to community variations for the bacterial
communities in permafrost soil, lake sediment, and lake water, respectively (Figure 8b). In
particular, the results showed that homogeneous selection contributed a larger fraction to
the assembly of the WBCs (44.8%), followed by SBCs (35.2%) and PBCs (29.7%) (Figure
8b). Heterogeneous selection influenced PBCs (21.6%) more than SBCs (6.0%) and WBCs
(0.1%). Dispersal limitation contributed a larger fraction to SBCs (57.2%) than to PBCs
(45%) and WBCs (37.5%).
The relationships between βNTI and major environmental variables were used to estimate
changes in the relative influences of deterministic and stochastic assembly processes.
Mantel tests showed that the assembly processes of bacterial communities in permafrost
soil, lake sediment, and lake water had similarities and differences in the responses to
environmental variables (Figure 7). Particularly, differences of TP were significantly
associated with βNTI of PBCs, SBCs, and WBCs, implying that an increasing divergence
of TP could contribute to a shift from homogeneous selection to heterogeneous selection
in the assembly of bacterial communities in the QTP thermokarst landscape. Moreover,
βNTI of PBCs was also significantly associated with other environmental variables, except
elevation, MAT, and conductivity. βNTI of SBCs was also significantly associated with
latitude, MAP, conductivity, pH, and C:N, while βNTI of WBCs was significantly
associated with longitude, conductivity, and TN.

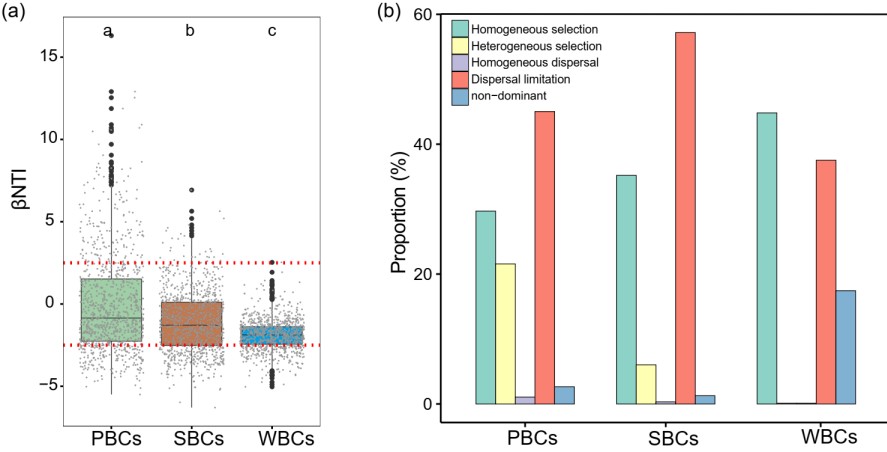


Figure 8 (a) The values of βNTI with horizontal dashed red lines indicate upper and lower
significance thresholds at βNTI = +2 and −2, respectively in the three habitat types. (b) The
contribution of deterministic (homogeneous and heterogeneous selection) and stochastic
(dispersal limitations and homogenizing dispersal) processes to turnover in the assembly
of PBCs, SBCs and WBCs. "Non-dominant" indicates that the fraction was not dominated
by any single process.
**4 Discussion**
Thermokarst lakes and degraded permafrost are distinct habitats derived from original
permafrost during the process of thermokarst formation. Degraded permafrost can be
further converted to thermokarst lake sediment during the continuous process of
thermokarst formation (Figure 1). In our studied thermokarst landscapes across the QTP,
bacterial communities in degraded permafrost soil (PCBs), thermokarst lake sediment
(SBCs), and thermokarst lake water (WCBs) differed in multiple aspects, such as α-
diversity, β-diversity, community composition, community assembly rules, and
environmental responses (Figure 1c), supporting a view in which thermokarst formation



generates novel habitat conditions and microbial communities in landscapes formerly
occupied by permafrost.

### 4.1 Alpha diversity and community composition

Permafrost soil and lake sediments on the QTP had significantly higher alpha diversity than
lake water. A considerable proportion (41%) of OTUs were shared among PCBs, SCBs,
and WCBs. However, besides a small number of unique OTUs, only a small proportion of
OTUs were enriched in lake water. In addition, bacterial communities were also
significantly different in composition and structure among permafrost soil, lake sediment,
and lake water, but with lower dissimilarities between PCBs and SBCs. Due to the origin
of thermokarst lakes from permafrost, there is no doubt that permafrost soil, lake sediments,
and lake water should share a certain number of OTUs.
Thermokarst lakes are known to have sediments that derive from the permafrost soil and
are constantly replenished by the collapse of nearby permafrost (Payette et al., 2004; West
and Plug, 2008; Veremeeva et al., 2021). This suggests that permafrost soil and lake
sediments are likely to have high levels of similarity in bacterial diversity and community
composition. Additionally, our prior research has shown that there are close correlations
between the abiotic features of the two environments (Ren et al., 2022b). However, despite
these similarities and connections, we found substantial differences in the bacterial
communities of permafrost soil and lake sediments. As proposed by the Baas-Becking
hypothesis (Baas-Becking, 1934), environmental selection is partially responsible for
variation in microbial communities, which are also shaped by other ecological processes,
such as diversification and dispersal limitation. Indeed, in our study, alpha diversity and



the dominant phyla found in PBCs and SBCs responded differently to various
environmental variables.
Bacterial communities in lake water had significantly lower alpha diversity as well as
distinct community composition and structure in comparison to bacterial communities in
permafrost soil and lake sediment. However, PBCs and SBCs had direct influence on
WBCs. For thermokarst lakes, the water first originates from the thawing of the ice-rich
permafrost and the lake is then fed by precipitation-derived and permafrost-derived water
(Yang et al., 2016a; Narancic et al., 2017; Wan et al., 2019). Microorganisms present in
lake water have a diverse range of sources, including terrestrial inputs and other sources
such as bacteria distributed with the atmosphere, associated with plants and animals, and
carried by migratory birds and animals (Ruiz-Gonzalez et al., 2015). Thus, there was a
relatively small proportion of OTUs shared between permafrost soil and lake water, as well
as between lake sediment and water, and only a few shared OTUs were enriched in lake
water. It is a well-established fact that different habitats often support distinct microbial
communities (Fierer et al., 2012; Hugerth et al., 2015; Louca et al., 2016). The contrast in
bacterial community composition between lake sediments and water has been extensively
documented (Briee et al., 2007; Gough and Stahl, 2011; Yang et al., 2016b; Ren et al.,
2017). In addition, sediment generally harbor a higher species-level diversity of bacteria
compared to lake water (Lozupone and Knight, 2007; Ren et al., 2019b). For example, in
a permafrost thaw pond of Andes, it was also found that water samples had lower alpha
diversity than lake sediment and permafrost samples (Aszalós et al., 2020). Permafrost soil
and lake sediment may provide more habitat heterogeneity for bacterial taxa than the water
column, supported by our observation that the bacterial taxa had higher niche breadth in





permafrost soil and lake sediment than in lake water. Moreover, in hydrologically
connected terrestrial-aquatic ecosystems, bacterial communities can present distinct but
directional spatial structure driven by terrestrial recruited taxa (Ruiz-Gonzalez et al., 2015).
Thus, these community similarities between distinct bacterial habitats might be the result
of common bacterial source (original permafrost) and the differences are likely caused by
subsequent environmental selection, colonization from multiple other bacterial sources,
and distinct assembly mechanisms.
Despite connections driven by dispersal, distinct thermokarst habitats had distinct bacterial
community composition, as seen in previous work (Ottoni et al., 2022). All the dominant
phyla were significantly different in relative abundance among permafrost soil (PBCs),
lake sediment (SBCs), and lake water (WBCs). In this study, Proteobacteria, Bacteroidota,
Actinobacteriota, Gemmatimonadota, Acidobacteriota, Desulfobacterota, and Chloroflexi
dominated bacterial communities in permafrost soil and/or thermokarst lakes despite high
variability. Similar dominance of these taxa has also been found in permafrost and
thermokarst landscapes in other areas (Aszalós et al., 2020; Belov et al., 2020; Wu et al.,
2022). The most commonly reported bacterial groups in permafrost environments include
members of Proteobacteria, Acidobacteria, Actinobacteria, Bacteroidetes, Firmicutes, and
Chloroflexi (Steven et al., 2009; Altshuler et al., 2017; Ottoni et al., 2022), as observed in
our samples.
***4.2 Beta diversity and assembly processes***
In our studied regions across the QTP, PBCs, SBCs, and WBCs all had a high beta diversity
(average values > 0.7), with WBCs showing the highest, suggesting that bacterial
communities shifted substantially across the large spatial scale of our sampling. Moreover,



beta diversities of PBCs, SBCs, and WBCs were significantly correlated with each other,
further suggesting that the bacteria in different habitats had a considerable proportion of
members from the same source, the original permafrost soil. The significantly lower mean
SES.MNTD for PBCs indicate that bacterial communities in permafrost soil were more
closely phylogenetically clustered than those in lake sediment and water (Langenheder et
al., 2017), consistent with the observation that PBCs had lower beta diversity than SBCs
and WBCs.
The structure of bacterial communities can vary across spatiotemporal scales and different
habitats (Ren et al., 2017; Aguilar and Sommaruga, 2020; Pearman et al., 2020). A key
objective in the field of microbial ecology is to determine the relative influence of
stochastic and deterministic processes in shaping the assembly of communities (Stegen et
al., 2013; Zhou and Ning, 2017). In this study, deterministic processes contributed 51.3%,
41.2%, and 44.9% to community variation for the bacterial communities in permafrost soil,
lake sediment, and lake water, respectively. Homogeneous selection contributed a larger
fraction to the assembly of the WBCs (44.8%), followed by SBCs (35.2%) and PBCs
(29.7%). Heterogeneous selection influenced PBCs (21.6%) more strongly than SBCs
(6.0%) and WBCs (0.1%). Dispersal limitation contributed a larger fraction to SBCs
(57.2%) than to PBCs (45%) and WBCs (37.5%). The dispersal of microorganisms is often
considered as a passive process that results in community variation and turnover coupled
with the function of environmental filtering (Cline and Zak, 2014; Stegen et al., 2015). The
high dispersal limitation of microbial communities in thermokarst lakes could be
potentially explained that the isolated nature of thermokarst lakes being endorheic results
in limited connectivity and strong restriction of microbial dispersal, as well as strong



environmental filtering. Additionally, the prolonged frozen phase of thermokarst lakes and
permafrost soil restrict the movement of microorganisms (Vargas Medrano, 2019;
Vigneron et al., 2019). Although the "everything is everywhere" hypothesis suggests that
many microorganisms have a cosmopolitan distribution, their slow mobility allows for the
development of regional phylogenetic differences and the emergence of specialized,
endemic taxa in isolated habitats, resulting in a low likelihood of microorganisms
dispersing to suitable distant sites (Telford et al., 2006). Therefore, dispersal processes in
this thermokarst landscape may be restricted by the lack of hydrological connection,
limited movement of water, short duration since thawing, and strong environmental
filtering, contributing to the observed high dispersal limitation in the studied permafrost
soil and thermokarst lakes. This inference is supported by many previous studies showing
that dispersal limitation plays a major role in structuring microbial communities in lakes
(Lindstrom and Langenheder, 2012; Yang et al., 2019; Liu et al., 2021). Strong dispersal
limitation for bacterial communities in permafrost has also been documented across an
Alaskan boreal forest landscape (Bottos et al., 2018). In addition, bacterial communities in
lake water displayed a higher influence of homogeneous selection compared to those in
lake sediments and permafrost soil in our study. The reason for this might be that long-
term changes in thermokarst lakes result in homogenized habitats and consequently strong
homogenous selection on bacterial communities (Ning et al., 2019). Deterministic
processes could also cause the communities to be more dissimilar through heterogeneous
selection, which also imposed strong control on PBCs.





### 4.3 Environmental influences


Understanding how environmental factors shape bacterial communities is a crucial aspect

in the field of microbial ecology (Fierer and Jackson, 2006; Pla-Rabes et al., 2011). With

global warming, climatic and physicochemical environments will be strongly altered in

permafrost areas. On the QTP in particular, air temperature and precipitation are increasing

in most regions (Xu et al., 2008; Lu et al., 2018). Moreover, organic carbon and nutrient

stocks in permafrost are decreasing (Turetsky et al., 2020; Wu et al., 2021) and thermokarst

lakes are developing, driving dynamic environmental change (Luo et al., 2015; Vucic et

al., 2020). These environmental disruptions likely impose strong influences on bacterial

communities in thermokarst landscapes. In our study, bacterial communities were

differentially correlated with various measured environmental variables. In our data, pH

was consistently identified as a strong correlate of microbial community structure and

diversity, as is often observed in terrestrial and aquatic ecosystems worldwide (Fierer and

Jackson, 2006; Xiong et al., 2012). While such correlations between pH and bacterial

communities have been widely found, the regulation mechanisms still remain unknown

(Malard and Pearce, 2018). Moreover, the influences of pH are often species- and location-

specific (Malard and Pearce, 2018; Egelberg et al., 2021). In this study, pH had

significantly negative relationships with alpha diversity of bacterial communities in lake

sediment and water, and had negative or positive correlations with some phyla. Moreover,

differences in pH might drive community variation observed between PBCs, SBCs, and

WBCs, and shift community assembly processes for PBCs and SBCs. Our study also

showed that nutrient (C, N, and P) concentrations and stoichiometric ratios were strongly

related to alpha diversity (particularly for SBCs) and community variation and assembly



(particularly for PBCs). The role of nutrient availability in shaping bacterial communities
has been well established (Torsvik et al., 2002; Lee et al., 2017; Zhou et al., 2020). High
organic matter content, for instance, has been shown to support diverse and complex
microbial communities (Garrido-Benavent et al., 2020; Ren and Gao, 2022). Due to their
ecological strategies, metabolic features, and environmental preferences, bacteria respond
differentially to nutrient status (Carbonero et al., 2014). Thermokarst lakes have sediment
directly formed from permafrost soil, and thus, permafrost soil environments and bacterial
communities had strong associations with that of lake sediment.
In addition to physicochemical environments, location and climate were also suggested to
influence bacterial communities in distinct habitats. Warming and altered precipitation
regimes under climate change have been demonstrated to affect alpha diversity and
composition of stream microbial communities at continental scales (Picazo et al., 2020).
Our study indicates that location (particularly latitude) and climate (particularly MAP)
factors are important in shifting bacterial communities in thermokarst landscapes.
Understanding large-scale pattern of bacterial communities is increasingly important to
offer insights into the impacts of climate change (Picazo et al., 2020; Ren et al., 2021). As
global climate changes, QTP is getting warmer and more humid (Xu et al., 2008; Lu et al.,
2018). Therefore, significant alterations to the physical, chemical, and biological properties
of thermokarst lakes on the QTP can be expected in the coming decades. Based on "space-
for-time" substitution, our study serves as a foundation for predicting the potential impact
of climate change on bacterial communities in thermokarst landscapes.



**5 Conclusion**
In this study, we investigated bacterial communities in paired water and sediment samples
in thermokarst lakes as well as permafrost soil around the lakes across the QTP. esults
showed that each habitat had distinct bacterial assemblages, with lower alpha diversity in
lake water and higher beta diversity in lake sediment and permafrost soil. There was
considerable overlap in OTUs across habitats. Bacterial communities in permafrost soil
and lake sediment were influenced by dispersal limitation, while those in lake water were
driven by homogeneous selection. Environmental variables, including latitude, mean
annual precipitation, and pH, affected bacterial community variations in all habitats. The
study highlights the unique bacterial communities and ecological impacts of permafrost
degradation in diverse habitats created by thermokarst processes.
**Acknowledgements**
We are grateful to Yongming Deng and Xuan Jia for their assistance in the field and
laboratory work.
**Data availability statement:**
Raw sequences were uploaded to the China National Center for Bioinformation
(PRJCA009850, CRA007082).
**Funding statement:**
This study was supported by the open funding of the State Key Laboratory of Lake Science
and Environment (2022SKL010).
**Conflict of interest disclosure:**





The authors declare no competing interests.
**Author's contributions:**
Z.R. designed the study, did the analyses, and prepared the manuscript, performed the field
work and laboratory work. All the authors prepared the manuscript.
**Ethics approval statement:**
Not applicable
**Permission to reproduce material from other sources:**
Not applicable
**Originality-Significance Statement:**
This is our original study and not submitted to elsewhere
**Supplementary Information**
Figure S1 Map of the 44 sampling sites of permafrost soil and thermokarst lakes across the
Qinghai-Tibet Plateau. The distribution of the permafrost was cited from Zou et al., 2017.
This map was cited from Ren et al, 2022a.
Figure S2 (a) Venn diagram showing the unique and shared OTUs among distinct bacterial
communities. (b) The volcano plot showing the shared OTUs that significantly (t-test, $P <$
0.05) enriched in a certain habitat. The volcano plot was constructed using $\log_2$ (fold
change) on x-axis and $-\log_{10}$ (p-values of t-test) on y-axis.
Figure S3 Distance-decay curves showing community similarity against geographic
distances between sampling sites. Solid lines denote the ordinary least-squares linear
regressions.





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
