# Peer review of "landscapes: implications for ecological consequences of permafrost"

_Biogeosciences, 2023_

## Author Comment (AC1)

Dear Editor and Reviewers,

Thank you for dedicating your time to review our manuscript and providing us with valuable feedback. We are grateful for the positive comments highlighting the potential significance and interest of our study. We highly value all the critical comments, as they have greatly contributed to the improvement of our work.

In response to the reviewer' comments, we have thoroughly revised the manuscript. We have carefully considered each comment and incorporated the necessary changes and refinements throughout the revised manuscript. To facilitate your review, we have provided detailed responses to each comment in blue color below. Additionally, you can refer to the tracked changes in the manuscript for a comprehensive overview of the revisions made.

Once again, we sincerely appreciate your time and expertise in evaluating our work, and we hope that the revisions have strengthened the manuscript in terms of clarity, accuracy, and overall quality.

Best Regards,

Ze Ren on behalf of all co-authors.

**RC1: 'Comment on bg-2023-85', Dajana Radujkovic, 29 Jun 2023**

**General comments:**

The study by Ren et al. investigates bacterial diversity and community composition as well as potential deterministic and stochastic processes that shape bacterial communities in three types of thermokarst habitats across the Qinghai-Tibet Plateau. The manuscript is clear and concise, the figures are informative and relevant, and overall the study is an important contribution to understanding bacterial community composition and assembly processes in thermokarst landscapes.

We sincerely appreciate your positive feedback on the potential importance of our study about bacterial diversity and community assembly in thermokarst habitats across the Qinghai-Tibet Plateau. We have carefully considered your comments and suggestions, and incorporated relevant revisions into the manuscript to further improve its clarity and scientific contribution. Thank you for your valuable feedback.

However, the manuscript would benefit from a more detailed explanation of certain analyses in the method section, given that some relevant information is lacking.

We appreciate your suggestion and understand the importance of providing sufficient detail on the analyses and methods. In response, we have revised the method section to include more explicit descriptions. Please referring to the responses below to the specific comments and the revised manuscript.

Moreover, the discussion could be expanded to give a more thorough interpretation of the results. Currenlty, the discussion is often very general and does not address some of the interesting findings of this study concretely. Below are more specific comments and examples.

We thank your comments on highlighting the need for a more comprehensive discussion that addresses the specific and interesting findings of our study. In response, we have expanded the discussion section to provide a more in-depth interpretation of the results. Please referring to the responses below to the specific comments and the revised manuscript.

**Specific comments:**

L139: The paragraph about PCR and sequencing is missing some important detail that would enable the reader to understand what exactly was done and why certain choices were made. First, even though PCR preparation, PCR conditions were described in a previous study, it would be good to describe them briefly in this paper. Moreover, could the authors describe other steps of library preparation, how were the PCR products cleaned and quantified? Were paired-end reads sequenced, and how many base pairs? The sequences were trimmed at the end; what was the length of the sequences, and what was the trim length? How were the low-quality sequences detected, and what was the threshold used? Which version of the Silva database was used (release date)? How were the sequences normalized, and why was this particular threshold used (24.251)?

Thanks for pointing out these omissions. We agree that providing a more comprehensive description of the PCR and sequencing protocols is necessary to ensure clarity and transparency. We have carefully revised the methods section to provide a more detailed description of the PCR and sequencing protocols.

For example, we added: "PCRs were conducted in 25 µl reaction mixture containing 2.5 µl of TransStart buffer, 2 µl of dNTPs, 1 µl of each primer, 0.5 µl of TransStart Taq DNA polymerase, and 20 ng template DNA. The PCR reactions were conducted on a thermal cycler (ABI GeneAmp® 9700, USA) using the followed procedure: initial denaturation at 94 °C for 5 min, 24 cycles of denaturation at 94 °C for 30 s followed by annealing at 56 °C for 30 s and extension at 72 °C for 20 s, and a final extension at 72 °C for 5 min.", "DNA libraries were verified on 2% agarose gels and quantified using a Qubit 4 Fluorometer

(Thermo Fisher Scientific, Waltham, USA).", "The sequences were subjected to the following denoising criteria: sequences with ambiguous or homologous regions, as well as those below 200 bp in length, were excluded; sequences with at least 75% of bases having a quality score above Q20 were retained; and chimeric sequences were identified and eliminated. All sequences from extraction blanks were removed.", "The effective sequences were grouped into Operational Taxonomic Units (OTUs) using a 97% sequence similarity threshold and annotated the taxonomic classifications against the SILVA 138 database (released on 02-Nov-2020)", "The singletons were removed, and the sequences were rarefied to the lowest number of sequences per sample (24,251 sequences) to eliminate the bias from the sampling effort."

Please referring to the revised manuscript for more details.

L155: Analyses: In some cases, multiple tests were performed (e.g. correlation tests). Did the authors apply any correction of P values for multiple testing?

We did the correction of P values using the FDR method (Benjamini and Hochberg, 1995). In the revision, we clarified in the methods section.

✧ Benjamini Y, Hochberg Y. 1995. Controlling the false discovery rate: a practical and powerful approach to multiple testing. Journal of the Royal Statistical Society: Series B (Methodological), 57: 289-300.

L176: Could the authors provide more details about the construction of SEM? How were the paths constructed, and why? The path construction should have some theoretical rationale. Did the authors assess the fit of SEM, and which parameters were used for this? Which package was used to construct SEM?

We apologize for the lack of clarity regarding the construction of the SEM in our initial manuscript. To address this concern, we have provided a more detailed explanation in the methods section regarding the construction of the paths in the SEM and the theoretical rationale behind them: "Structural equation modeling (SEM) was conducted to assess the relationships among location (including latitude, longitude, and elevation), climate (including mean annual temperature and mean annual precipitation), and physicochemical variables (including pH, conductivity, nutrients concentrations and stoichiometric ratios) of each habitat (permafrost soil, lake sediment, and lake water), as well as their bacterial communities (PBCs, SBCs, and WBCs). In model building, the SEM incorporated prior knowledges: (a) location and climate factors potentially influence all the studied bacterial communities, (b) physicochemical factor of each habitat potentially influences the corresponding bacterial communities, and (c) permafrost soil potentially influences thermokarst lake sediment and water, while lake sediment and water interact with each other."

In our study, we did assess the fit of the SEM. In the revision, we clarified in the methods that: "SEM was constructed using the lavaan package (Rosseel, 2012). The fit of SEM was assessed using standard indices, including chi-square ($\chi 2$), goodness-of-fit index (GFI), comparative fit index (CFI), root mean square residual (RMR), and root mean squared error of approximation (RMSEA) (Hu and Bentler, 1999; Barrett, 2007)." In the figure 5b of SEM, we added "$\chi 2 = 37.867$, df = 11, GFI = 0.913, CFI = 0.867, RMR = 0.269, RMSEA = 0.023"

✧ Barrett, P.: Structural equation modelling: Adjudging model fit, Pers. Individ. Differ., 42, 815-824, doi:10.1016/j.paid.2006.09.018, 2007.

✧ Hu, L. and Bentler, P. M.: Cutoff criteria for fit indexes in covariance structure analysis: Conventional criteria versus new alternatives, Structural equation modeling, 6, 1-55, doi:10.1080/10705519909540118, 1999.

✧ Rosseel, Y.: lavaan: An R Package for Structural Equation Modeling, J. Stat. Softw., 48, 1-36, doi:10.18637/jss.v048.i02, 2012.

L348-351: Could the authors explain in more detail how this analysis was performed? E.g. how were homogeneous and heterogeneous selection determined?

In the methods section, we clarified as: $\beta$NTI values $<-2$ or $>+2$ indicate signals for heterogeneous selection and homogenous selection, respectively. The values with $-2< \beta$NTI $<2$ and RC$_{Bray}$ $<-0.95$ indicate homogeneous dispersal, while $-2< \beta$NTI $<2$ and RC$_{Bray}$ $>0.95$ indicate dispersal limitation. The values with $-2< \beta$NTI $<2$ and $-0.95< $ RC$_{Bray}$ $<0.95$ indicate "undominated".

L433-435: Could the authors provide some possible explanations for this?

In the revision, we added more possibilitble explanations: "The significantly lower mean SES.MNTD for PBCs indicate that bacterial communities in permafrost soil were more closely phylogenetically clustered and suffered stronger environmental filtering than those in lake sediment and water (Langenheder et al., 2017), consistent with the observation that PBCs had lower beta diversity than SBCs and WBCs. SES.MNTD is sensitive to changes in lineage close to the phylogenetic tips (Kembel et al., 2010). The higher SES.MNTD observed for SBCs and WBCs suggest the possibility that the bacteria in lake sediment and water exhibit a substantial divergence in the co-occurring species, and thermokarst lakes have experienced colonization by bacterial species originating from distinct clades or lineages from external sources following permafrost thaw (Webb et al., 2002; Stegen et al., 2013)."

✧ Kembel, S. W., Cowan, P. D., Helmus, M. R., Cornwell, W. K., Morlon, H., Ackerly, D. D., Blomberg, S. P. and Webb, C. O.: Picante: R tools for integrating phylogenies and ecology, Bioinformatics, 26, 1463-1464, doi:10.1093/bioinformatics/btq166, 2010.

✧ Langenheder, S., Wang, J., Karjalainen, S. M., Laamanen, T. M., Tolonen, K. T., Vilmi, A. and Heino, J.: Bacterial metacommunity organization in a highly connected aquatic system, FEMS Microbiol. Ecol., 93, fiw225, doi:10.1093/femsec/fiw225, 2017.

✧ Stegen, J. C., Lin, X., Fredrickson, J. K., Chen, X., Kennedy, D. W., Murray, C. J., Rockhold, M. L. and Konopka, A.: Quantifying community assembly processes and identifying features that impose them, The ISME Journal, 7, 2069-2079, doi:10.1038/ismej.2013.93, 2013.

✧ Webb, C. O., Ackerly, D. D., Mcpeek, M. A. and Donoghue, M. J.: Phylogenies and community ecology, Annual review of ecology and systematics, 33, 475-505,2002.

L441-447: Instead of repeating the results in detail, it would perhaps be more useful to focus on discussing these results and interpreting what they mean for each habitat. For instance, the general discussion about dispersal limitation was interesting, but more specifically for the study, it would be interesting to discuss, e.g. why heterogeneous selection influenced PBCs much more strongly than WBCs and why dispersal limitation was the most important for sediments.

In the revision, we deleted some detail description of the results. In addition, we added more discussions about the differences of community assemblage. For example, we added: "Long-term changes in thermokarst lakes result in homogenized habitats and consequently strong homogenous selection on bacterial communities (Ning et al., 2019). In contrast, permafrost soil is a highly heterogeneous environment across spatial scales (Etzelmüller, 2013; Nitzbon et al., 2021), creating a wide range of habitats which can impose strong heterogeneous selection pressures on bacterial communities. Furthermore, permafrost soil is characterized by limited nutrient availability due to the frozen state of organic matters (Beermann et al., 2017; Zhang et al., 2023), while lake water offers a more diverse and abundant array of dissolved organic compounds and nutrients. As a result, bacterial communities in permafrost soil might be more sensitive to variations in resource availability, rendering them more strongly influenced by heterogeneous selection", "Furthermore, geographical barriers, exemplified by prominent mountain ranges like the Tanggula Mountains, Kunlun Mountains, Nyenchen Tanglha Mountains, and Bayan Har Mountains, serve as impediments to the dispersal of both macro- and microorganisms (Wan et al., 2016; Yu et al., 2019; Ren et al., 2022c)", and "Particularly in lake sediment, where bacterial communities are more isolated over distances and will not disperse as far as those in lake water and permafrost soil, resulting in strong influence of dispersal limitation (Martiny et al., 2006; Xiong et al., 2012)."

✧ Beermann, F., Langer, M., Wetterich, S., Strauss, J., Boike, J., Fiencke, C., Schirrmeister, L., Pfeiffer, E. M. and Kutzbach, L.: Permafrost thaw and liberation of inorganic nitrogen in Eastern Siberia, Permafrost and Periglacial Processes, 28, 605-618,2017.

✧ Etzelmüller, B.: Recent advances in mountain permafrost research, Permafrost and Periglacial Processes, 24, 99-107,2013.

✧ Martiny, J., Bohannan, B., Brown, J. H., Colwell, R. K., Fuhrman, J. A., Green, J. L., Horner-Devine, M. C., Kane, M., Krumins, J. A., Kuske, C. R., Morin, P. J., Naeem, S., Ovreas, L., Reysenbach, A. L., Smith, V. H. and Staley, J. T.: Microbial biogeography: putting microorganisms on the map, Nat. Rev. Microbiol., 4, 102-112, doi:10.1038/nrmicro1341, 2006.

✧ Ning, D., Deng, Y., Tiedje, J. M. and Zhou, J.: A general framework for quantitatively assessing ecological stochasticity, Proceedings of the National Academy of Sciences, 116, 16892-16898, doi:10.1073/pnas.1904623116, 2019.

✧ Nitzbon, J., Langer, M., Martin, L. C. P., Westermann, S., Schneider Von Deimling, T. and Boike, J.: Effects of multi-scale heterogeneity on the simulated evolution of ice-rich permafrost lowlands under a warming climate, The cryosphere, 15, 1399-1422, doi:10.5194/tc-15-1399-2021, 2021.

✧ Ren, Z., Jia, X., Zhang, Y. T., Ma, K., Zhang, C. and Li, X.: Biogeography and environmental drivers of zooplankton communities in permafrost-affected lakes on the Qinghai-Tibet Plateau, Glob. Ecol. Conserv., 38, e02191, doi:10.1016/j.gecco.2022.e02191, 2022.

✧ Wan, D. S., Feng, J. J., Jiang, D. C., Mao, K. S., Duan, Y. W., Miehe, G. and Opgenoorth, L.: The Quaternary evolutionary history, potential distribution dynamics, and conservation implications for a Qinghai-Tibet Plateau endemic herbaceous perennial, Anisodus tanguticus (Solanaceae), Ecol. Evol., 6, 1977-95, doi:10.1002/ece3.2019, 2016.

✧ Xiong, J., Liu, Y., Lin, X., Zhang, H., Zeng, J., Hou, J., Yang, Y., Yao, T., Knight, R. and Chu, H.: Geographic distance and pH drive bacterial distribution in alkaline lake sediments across Tibetan Plateau, Environ. Microbiol., 14, 2457-2466, doi:10.1111/j.1462-2920.2012.02799.x, 2012.

✧ Yu, H., Favre, A., Sui, X., Chen, Z., Qi, W., Xie, G., Kleunen, M. and van Kleunen, M.: Mapping the genetic patterns of plants in the region of the Qinghai–Tibet Plateau: Implications for conservation strategies, Diversity & distributions, 25, 310-324, doi:10.1111/ddi.12847, 2019.

✧ Zhang, D., Wang, L., Qin, S., Kou, D., Wang, S., Zheng, Z., Peñuelas, J. and Yang, Y.: Microbial nitrogen and phosphorus co‑limitation across permafrost region, Glob. Change Biol., 29, 3910-3923, doi:10.1111/gcb.16743, 2023.

L473: Could this be discussed further?

We revised by reorganizing this paragraph and adding more discussion on the differences of bacterial community assemblage in lake water, sediment, and permafrost soil. Please referring to the previous response.

L494-504: The authors write about the general importance of different environmental factors for bacterial communities but do not go into why certain of these factors were important in this study. Could the authors discuss more specifically why particular environmental factors are important in these different habitats and, even more interestingly, why the environment seems to be more important in PBCs than the other habitats? For instance, in this respect, it could be interesting to discuss the results of SEM.

In the revision, we added more specific discussions: "Compared to permafrost soil and lake water, lake sediment can exhibit more stable physicochemical conditions. However, permafrost soil and lake water experience more dynamic and extreme environmental

changes, which drive the bacterial communities. The results of SEM also in line with bacterial community assembly that deterministic processes had stronger influences on PBCs and WBCs than on SBCs", "Particularly for bacterial communities in permafrost soil, location and climate have been evidenced as strong factors in shaping microbial communities (Taş et al., 2018; Barbato et al., 2022)."

✧ Barbato, R. A., Jones, R. M., Douglas, T. A., Doherty, S. J., Messan, K., Foley, K. L., Perkins, E. J., Thurston, A. K. and Garcia-Reyero, N.: Not all permafrost microbiomes are created equal: Influence of permafrost thaw on the soil microbiome in a laboratory incubation study, Soil Biology and Biochemistry, 167, 108605, doi:https://doi.org/10.1016/j.soilbio.2022.108605, 2022.

✧ Taş, N., Prestat, E., Wang, S., Wu, Y., Ulrich, C., Kneafsey, T., Tringe, S. G., Torn, M. S., Hubbard, S. S., Jansson, J. K., Pacific Northwest National Laboratory Pnnl, R. W. U. S. and Lawrence Berkeley National Laboratory Lbnl, B. C. U. S.: Landscape topography structures the soil microbiome in arctic polygonal tundra, Nat. Commun., 9, 777-13, doi:10.1038/s41467-018-03089-z, 2018.

---

## Author Comment (AC3)

Dear Editor and Reviewer,

We extend our gratitude for your dedicated time invested in reviewing our manuscript and for sharing your invaluable insights. Your positive feedback is deeply appreciated. Your constructive comments have significantly contributed to the enhancement of our work.

According to your comments, we have thoroughly revised the manuscript. We incorporated the necessary changes and refinements throughout the revised manuscript. To facilitate your review, we have provided detailed responses to each comment in blue color below. Additionally, you can refer to the tracked changes in the manuscript for a comprehensive overview of the revisions made.

Once again, we sincerely acknowledge the value of your time and your expert evaluation of our work. We hope that these revisions have fortified the manuscript's lucidity, precision, and overall quality.

Best Regards,

Ze Ren on behalf of all co-authors.

**RC2: 'Comment on bg-2023-85', Anonymous Referee #2, 28 Aug 2023 reply**

In the study by Ren et al, the authors investigate microbial community assemblages in different degraded environments, degraded permafrost soils, thermokarst lake sediments and lake water, with the aim to identify dispersion and assembly processes. Although the communities differed among the environments, they nevertheless shared 41% of OTUs which suggests that and taxa disperse among the systems.

The manuscript is very clearly structured and well written, and the introduction provides a good overview of the topic. Moreover, due to the increasing possibility of enhanced thermokarst lake formation the authors elucidate and highlight microbial colonization pathways of those newly formed ecosystems.

We greatly appreciate your positive comments and constructive suggestions regarding our study. Your comments and suggestions have been meticulously reviewed and integrated into the manuscript to improve its quality. Please refer to the responses below and revisions in the revised manuscript for details.

As pointed out by reviewer 1, more information on DNA extraction and sequencing should be included, as well as on the assumptions made in the base-SEM. Moreover, the rationale for assembly processes could be explained a more detailed, especially the definitions of homogenous and heterogenous selection and how their contribution, and dispersal

limitation were estimated, as well as how deterministic and stochastic processes (and what deterministic processes would that be) were defined? Could this maybe elaborated more in the introduction already

For DNA extraction and sequencing, more details are provided in the revised manuscript, such as: "The Magen Hipure Soil DNA Kit (Magen, China) was used to extract DNA from soil (0.5 g frozen soil), sediment (0.5 g frozen sediment), and water (membrane filter) samples according to the manufacturer's protocols. Extraction blanks were routinely performed in parallel.", "Next generation sequencing of the amplicon products was conducted on an Illumina Miseq Platform (Illumina, San Diego, CA, USA). Automated cluster generation and 250/300 paired-end sequencing with dual reads were performed following the manufacturer's instructions.". We also added more details about the PCR and sequence processing. Please refer to the revised manuscript for detail.

For the assumptions in SEM, we added: "In model building, the SEM incorporated prior knowledges: (a) location and climate factors potentially influence all the studied bacterial communities, (b) physicochemical factor of each habitat potentially influences the corresponding bacterial communities, and (c) permafrost soil potentially influences thermokarst lake sediment and water, while lake sediment and water interact with each other."

For assembly processes, we added more details in the METHODS section: "Because homogeneous selection results in communities that share greater phylogenetic similarity, the proportion of homogeneous selection was calculated as the fraction of pairwise comparisons with $\beta NTI < -2$. On the other hand, heterogeneous selection, leading to communities with lesser phylogenetic similarity, was measured as the fraction of pairwise comparisons with $\beta NTI > +2$. Because homogeneous dispersal results in communities exhibiting greater taxonomic resemblance, the extent of its impact was measured as the proportion of pairwise comparisons with $-2 < \beta NTI < 2$ and $RC_{Bray} < -0.95$. Conversely, communities constrained by dispersal limitation display lesser taxonomic similarity, and the measure of dispersal limitation was derived from the fraction of pairwise comparisons with $-2 < \beta NTI < 2$ and $RC_{Bray} > 0.95$. Finally, the fraction of the pairwise comparisons with $-2 < \beta NTI < 2$ and $-0.95 < RC_{Bray} < 0.95$ was identified as "undominated".

We have had the explanation of deterministic and stochastic processes in the INTRODUCTION.

The discussion still has large stretches with results being repeated, which could be streamlined and put into a larger context, by relating to other findings.

In the revision, we have deleted the repeated information. In addition, we added more discussion by relating to other studies. Most of the revisions were made in the 4.2 and 4.3 sections. Please refer to the revised manuscript for detail.

**More specific comments:**

Across the manuscript: the abbreviations for the three studied ecosystems are not very intuitive (Permafrost soil and lake sediments are PCBs and SCBs), maybe they could be simplified?

Thanks for this great suggestion, we have simplified the bacterial communities in permafrost soil, lake sediment, and lake water as PB, SB, WB, respectively.

Fig. 1 the letters in particular in Fig. 1a and b are very small and hard to read, please increase size.

We made the revision on this figure and others figures with the same issue.

Fig. 6: How exactly was habitat niche breadth determined (based on OTU distribution?) and I am wondering if maybe Fig. 5 and 6 be merged into one, as they seem a bit redundant (as also the nmds is displaying the Bray Curtis distances, if I understood correctly.

Thanks for this suggestion. We combined Fig 5 and 6 together.

For the niche breadth, we added more details in the METHODS section: "In order to determine the habitat niche occupied by each taxon, we utilized the "spaa" package (Zhang, 2016) in R to calculate the Levin's niche width (Levins, 1968). The formula of niche breadth is $B_i = 1/\sum_1^n p_i^2$, where $B_i$ represents the niche breadth of OTU$_i$ across the communities, $n$ is the total number of communities, and $p_i$ is the proportion of OTU$_i$ in each community.". In the METHODS, we also clarified that: "The NMDS was based on the Bray-Curtis distance using the relative abundance of OTUs."

Line 501 and following lines: this is a very general statement, are there any studies that could here focus more on bacterial differences in thermokarst lakes, or at least in cold/permafrost ecosystems?

In the revision, we reorganized the whole section, "4.3 Environmental influences". We added some statement by citing other studies about permafrost microbes, such as Mackelprang et al, 2017; Romanowicz and Kling, 2022; Fu et al, 2023. Please refer to the revised manuscript for detail.

"For example, Actinobacteria and Gemmatimonadota have a negative, while Gemmatimonadota has a positive relationship with organic carbon and nutrients in

permafrost (Romanowicz and Kling, 2022; Fu et al, 2023), in line with our results. The fact that different bacterial phyla exhibited varied responses to changes in organic carbon and nutrient further emphasizes the intricate interplay between microorganisms and their environment. Due to their ecological strategies, metabolic features, and environmental preferences, bacteria in permafrost respond differentially to nutrient status and other stressors, driving adaptive changes in community composition and function (Mackelprang et al, 2017)."

✧ Fu L, Xie R, Ma D, Zhang M, Liu L. 2023. Variations in soil microbial community structure and extracellular enzymatic activities along a forest‐wetland ecotone in high‐latitude permafrost regions. Ecology and Evolution, 13: e10205-n/a.

✧ Mackelprang R, Burkert A, Haw M, Mahendrarajah T, Conaway CH, Douglas TA, et al. 2017. Microbial survival strategies in ancient permafrost: insights from metagenomics. The Isme Journal, 11: 2305-2318.

✧ Romanowicz KJ, Kling GW. 2022. Summer thaw duration is a strong predictor of the soil microbiome and its response to permafrost thaw in arctic tundra. Environmental Microbiology, 24: 6220-6237.

---

## Author Response (AR2)

Dear Editor,

Thank you for dedicating your time on our manuscript and your positive comments on our work and the revisions.

For your comment on Line 143: please add the water volume filtered to get sufficient amount of DNA. Here is our revision.

In Line 117-121, we clarified as: "For water sampling of each lake, surface water samples were collected at a depth of 0.3 to 0.5 m in three acid-clean bottles. For microbial analysis, 200 mL of water from each bottle was filtered using a 0.2-μm polycarbonate membrane filter (Whatman, UK) for DNA extraction. For each lake, three filters were combined into one composite sample for DNA extraction."

In Line 143, we clarified as: "…, and water (membrane filters by filtering 600 mL lake water in total) samples according to the manufacturer's protocols."

Thanks again for your time and expertise in evaluating our work.

Best Regards,

Ze Ren on behalf of all co-authors.